# The Cyclic Imine Core Common to the Marine Macrocyclic Toxins Is Sufficient to Dictate Nicotinic Acetylcholine Receptor Antagonism

**DOI:** 10.3390/md22040149

**Published:** 2024-03-27

**Authors:** Yves Bourne, Gerlind Sulzenbacher, Laurent Chabaud, Rómulo Aráoz, Zoran Radić, Sandrine Conrod, Palmer Taylor, Catherine Guillou, Jordi Molgó, Pascale Marchot

**Affiliations:** 1Lab “Architecture et Fonction des Macromolécules Biologiques” (AFMB), Aix-Marseille Univ, CNRS, Faculté des Sciences Campus Luminy, 13288 Marseille cedex 09, France; yves.bourne@univ-amu.fr (Y.B.); gerlind.sulzenbacher@univ-amu.fr (G.S.); 2Institut de Chimie des Substances Naturelles (ICSN), Univ Paris-Saclay, CNRS, 91198 Gif-sur-Yvette, France; laurent.chabaud@u-bordeaux.fr (L.C.); catherine.guillou@cnrs.fr (C.G.); 3Service d’Ingénierie Moléculaire pour la Santé (SIMoS) EMR CNRS 9004, Département Médicaments et Technologies pour la Santé, Institut des Sciences du Vivant Frédéric Joliot, CEA, INRAE, Université Paris-Saclay, 91191 Gif-sur-Yvette, France; romulo.araoz@cea.fr (R.A.); jordi.molgo@cea.fr (J.M.); 4Skaggs School of Pharmacy and Pharmaceutical Sciences (SSPPS), University of California San Diego, La Jolla, CA 92093-0751, USA; zradic@health.ucsd.edu (Z.R.); pwtaylor@health.ucsd.edu (P.T.); 5Centre de Recherche en Neurobiologie et Neurophysiologie de Marseille (CRN2M), Aix Marseille Univ, CNRS, 13344 Marseille, France; conrod@cerege.fr

**Keywords:** acetylcholine-binding protein, binding affinity, competitive antagonism, crystal structure, cyclic imine, electrophysiology, nicotinic acetylcholine receptor, pharmacophore, receptor subtype selectivity, spiroimine

## Abstract

Macrocyclic imine phycotoxins are an emerging class of chemical compounds associated with harmful algal blooms and shellfish toxicity. Earlier binding and electrophysiology experiments on nAChR subtypes and their soluble AChBP surrogates evidenced common trends for substantial antagonism, binding affinities, and receptor-subtype selectivity. Earlier, complementary crystal structures of AChBP complexes showed that common determinants within the binding nest at each subunit interface confer high-affinity toxin binding, while distinctive determinants from the flexible loop C, and either capping the nest or extending toward peripheral subsites, dictate broad versus narrow receptor subtype selectivity. From these data, small spiroimine enantiomers mimicking the functional core motif of phycotoxins were chemically synthesized and characterized. Voltage-clamp analyses involving three nAChR subtypes revealed preserved antagonism for both enantiomers, despite lower subtype specificity and binding affinities associated with faster reversibility compared with their macrocyclic relatives. Binding and structural analyses involving two AChBPs pointed to modest affinities and positional variability of the spiroimines, along with a range of AChBP loop-C conformations denoting a prevalence of antagonistic properties. These data highlight the major contribution of the spiroimine core to binding within the nAChR nest and confirm the need for an extended interaction network as established by the macrocyclic toxins to define high affinities and marked subtype specificity. This study identifies a minimal set of functional pharmacophores and binding determinants as templates for designing new antagonists targeting disease-associated nAChR subtypes.

## 1. Introduction

Cyclic imine toxins are a globally distributed and emerging family of marine macrocyclic biotoxins comprising seven different main groups of low-molecular-weight organic compounds: the gymnodimines, spirolides, pinnatoxins, pteriatoxins, portimines, prorocentrolides, and spiroprorocentrimine (for reviews, see [1,2,3,4,5]). Even though most of the discovered toxins with a characteristic cyclic imine function were found primarily in shellfish, most of the available evidence strongly indicates that marine dinoflagellates are responsible for the production of cyclic imine toxins. These lipophilic toxins as well as a large number of acyl derivative products of shellfish metabolism [6,7,8,9,10,11] can bioaccumulate and contaminate filter-feeding (bivalves) edible mollusks and other marine invertebrate species [12]. Therefore, they represent a risk for shellfish consumers (for reviews, see [13,14]), especially since some emerging cyclic imine toxins have the ability to cross the intestinal, blood–brain, and placental barriers [15].

The *Karenia selliformis* dinoflagellate produces gymnodimines A–C [16,17,18,19,20], while 12-methyl gymnodimine-A, 12-methyl gymnodimine-B, gymnodimine-D, and 16-desmethyl gymnodimine-D are produced by the dinoflagellate *Alexandrium ostenfeldii* [21,22,23,24,25,26]. The use of liquid chromatography-high resolution mass spectrometry has considerably expanded the number of known gymnodimine congeners (gymnodimines F–J) [11]. The dinoflagellate *A. ostenfeldii* has primarily been associated with the production of spirolides [27,28], which constitute the largest and most highly diverse group among the cyclic imine toxins [29,30,31,32]. The cosmopolitan benthic dinoflagellate *Vulcanodinium rugosum* [33,34,35] is the producer of pinnatoxins [36,37,38], portimines A and B [39,40], and kabirimine [41]. Pteriatoxins A–C, first isolated from the Okinawan oyster *Pteria penguin* [42], are supposed to be shellfish metabolites of a cyclic imine toxin precursor [43]. The dinoflagellate origin of pteriatoxins remains to be determined. Finally, *Prorocentrum lima* and *P. maculosum* have been related to the biosynthesis of prorocentrolides A and B, respectively [44,45,46], of which three analogs, prorocentrolide C and 4-hydroxyprorocentrolide from the benthic dinoflagellate *P. lima* [47], and spiro-prorocentrimine from a benthic *Prorocentrum* sp. of Taiwan [30,48], were also isolated. 

The chemical structures of the cyclic imine toxins exhibit a rich diversity involving a macrocyclic ring (14 to 27 carbon atoms) and two constant moieties: the cyclic imine (mostly found as a spiroimine) and the spiroketal ring system. In prorocentrolides, the 26-membered carbo-macrocycle or 28-membered macrocyclic lactone is arranged around a hexahydroisoquinoline that incorporates the cyclic imine group [43,44]. In turn, the cyclic imines are composed of 5-membered (portimines), 6-membered (gymnodimines, spiroprorocentrimine, prorocentrolides, and kabirimine), or 7-membered rings (spirolides, pinnatoxins, pteriatoxins). The other structural constituents of the ring system are one or two tetrahydrofurans (in portimine and gymnodimine A, and in gymnodimine D, respectively), or a tetrahydropyran (in prorocentrolides and spiroprorocentrimine), or more complex 6,5-(spirolides H and I), 6,6,5-(spirolide G), 6,5,5-(spirolides A–F), or 6,5,6-spiroketal moieties (in pinnatoxins and pteriatoxins) (for reviews, see [3,30,49]).

Most, if not all, of these toxins have been identified as competitive antagonists of the nicotinic acetylcholine receptors (nAChRs), initially from the central neurological symptoms and rapid lethality that they induced in mice, and then through the recording of nicotinic currents from various nAChR subtypes (for reviews, see [3,50,51]). The nAChRs are prototypical cation-selective, ligand-gated ion channels (LGIC) that mediate fast neurotransmission in the central and peripheral nervous systems [52,53]. They belong to the Cys-loop subfamily of LGICs and are formed by distinct combinations of five subunits that confer selectivity in pharmacological properties and regional tissue locations. In mammals, the diversity in the nAChR subunit subtypes and assemblies is most evident in the central nervous system, where up to nine α and three β subunits have been described, of which some can arrange as homopentamers with five acetylcholine (ACh)-binding sites (e.g., α7 and α9 subtypes) or as heteropentamers of two different subunits with either two or three ACh-binding sites (e.g., α3β2 and α4β2 subtypes) [54,55,56]. An additional level of complexity arises from the assembly of four distinct subunits in the muscle-type nAChR, α1_2_βγδ [54,56]. The various nAChRs used in this study are further described in Appendix B, and their subunit sequences are displayed in Figure A1.

During the last decade, an impressive number of medium- to high-resolution structures of several full-length LGICs in the absence or presence of bound ligands have been solved by X-ray crystallography or cryo-electron microscopy (EM) (for reviews, see [55,56,57,58]) including those of the α7 [59], α4β2 [60], and α1_2_βγδ nAChR subtypes [61]. These structures provided fundamental insights into the gating mechanisms of LGICs. However, despite the remarkable technological and experimental progress, using full-length receptors for identifying the molecular determinants involved in agonist and competitive antagonist binding remains both time and cost consuming, and risky in terms of success. 

The soluble ACh-binding proteins (AChBP) from mollusks form homopentameric assemblies of subunits homologous to the N-terminal, extracellular ligand binding domain (LBD) of the nAChR [62,63,64] (Figure A1). In addition to the overall structural features of the subunits, the aromatic side chains that form the ligand-binding pocket at the subunit interfaces are well conserved in the nAChR family, with greater variability for residues at the complementary or (−) face than the principal or (+) face of each interface. The binding pocket of AChBP possesses all of the functional residues identified in the nAChR LBD, and its extension toward various directions of the interface provides multiple means for selective accommodation of the nicotinic ligands [63,64,65,66,67,68]. Overall, nicotinic full and partial agonists recognize the “core agonist signature motif” central to the binding pocket and capped by loop C, located on the (+) face, whereas the larger competitive antagonists also extend toward peripheral directions along the interface, resulting in the opening of loop C and often in greater subtype selectivity than seen for agonists (for recent reviews, see [69,70]). However, AChBPs are devoid of the transmembrane domains and intracellular loops typical for nAChRs, and hence, of the capacity of conducting ions and mediating neurotransmission.

Previously, we documented the parameters and mode of binding of two pairs of closely related macrocyclic imine toxins, 13-desmethyl spirolide C (SPX) and (−)-gymnodimine-A (GYM) [71], and pinnatoxins A (PnTxA) and G (PnTxG) [72] (Figure 1) to several representative nAChR subtypes and two AChBP subtypes. Our data identified the molecular determinants on both the toxins and receptors dictating potent nicotinic antagonism. They also identified those responsible for the broad selectivity of SPX and GYM toward the various nAChR subtypes, and for the narrow selectivity of the pinnatoxins toward the muscle-type α1_2_βγδ or neuronal α7 nAChR subtypes *versus* the neuronal α3β2 and α4β2 nAChRs. In particular, we showed that affinity is dictated by the protonated imine nitrogen common to the macrocyclic toxins, while specificity is imposed by toxin determinants extending out of the agonist-binding nest toward apical, radial, or ‘membrane’ sides of the LBD. Since peptidic neurotoxins acting as subtype-selective nAChR antagonists (e.g., curarimimetic α-neurotoxin, waglerins, α-conotoxins) are polar molecules unable to cross the blood–brain barrier, we propose that the lipophilic macrocyclic imine framework might offer new avenues for distinguishing nAChR subtype functions in the brain.

A synthetic tetrahydrofuran fragment mimicking the C10–C20 skeleton of GYM was found not to bind the α1_2_βγδ nAChR [73]. In contrast, a series of small synthetic analogs of its 6,6-spiroimine core were found to inhibit ACh-evoked nicotinic currents on the α4β2 and α1_2_βγδ nAChR subtypes, although they were much less active than the parental phycotoxin. These data revealed that the spiroimine moiety is critical for the blockade of nAChRs and pointed to it as a possible pharmacophore of this group of toxins [74]. To confirm the identity of the minimal core motif dictating nAChR antagonism, we then synthesized a new analog of 6,6-spiroimine, differing from the previous ones through incorporation of a dioxolane moiety. This molecule was first generated as a (±) racemate and named “spiroimine” [75].

Here, from this racemic spiroimine, we purified and characterized the two enantiomers, (+) **R** and (−) **S**, and carried out a comprehensive study of their mode of action by recording voltage-clamp currents from the muscle-type α1_2_βγδ and neuronal α7 and α4β2 nAChRs along with binding parameters on AChBPs from *Aplysia californica* (A-AChBP) and *Lymnaea stagnalis* (L-AChBP), and by solving X-ray structures of their A-AChBP complexes. This study identified a minimal set of functional determinants and binding sites as a framework for the design of new effectors targeting disease-associated nAChR subtypes.

## 2. Results and Discussion

### 2.1. Chemical Synthesis and Characterization of the Spiroimine Enantiomers

The spiroimine (±)-**4** racemate was synthesized in three steps from diketone **1** using standard procedures (Figure 1; Appendix C). Separation by chiral HPLC of the two enantiomers present in (±)-**4** yielded virtually pure spiroimines (+)-**4** and (−)-**4**.

To assign the absolute configuration of the enantiopure spiroimines, (−)-**4** was also synthesized from ketone (+)-**5 S**, prepared according to [76]. Spiroimine (−)-**4** was found to have configuration **S** at the quaternary carbon (Figure 2; Appendix C), and hence, spiroimine (+)-**4** was considered to be of the opposite configuration **R**. The two enantiomers were therefore designated as (+)-**4 R** and (−)-**4 S**.

Initial electrophysiology experiments on nAChRs and binding and crystallography experiments on AChBPs were carried out using the racemic spiroimine (±)-**4**. These were then complemented by more comprehensive experiments performed with the enantiopure spiroimines (+)-**4 R** and (−)-**4 S**. 

### 2.2. Functional Characteristics and nAChR Subtype Selectivity of the Spiroimine Racemate and Enantiomers

Functional analysis of the (±)-**4** racemate and (+)-**4 R** and (−)-**4 S** enantiomers used either manual or automated two-electrode voltage-clamp (TEVC) recordings on *Xenopus laevis* oocytes at −60 mV holding membrane potential. The oocytes were either micro-transplanted with *Torpedo* α1_2_βγδ-enriched electrocyte membranes or transfected with human α7 or α4β2-encoding cDNAs. When applied alone (see *Experimental Procedures*), none of the three spiroimines induced nAChR channel opening, thereby discarding any agonistic activity toward these nAChR subtypes. Instead, when applied together with ACh at its EC_50_ concentration for each nAChR subtype, all three spiroimines clearly behaved as nicotinic antagonists, whose potency depended on the receptor subtype examined (Figure 2; Table 1).

Indeed, the concentration–response curves recorded on the α1_2_βγδ nAChR showed a full antagonistic effect at high spiroimine concentration, with IC_50_ values in the *low* µM range for the three spiroimines, with no stereospecificity and no apparent cooperativity or difference in the binding affinities to the αγ and αδ interfaces (Figure 2a *left*; Table 1). Perfusion of a mixed ACh/spiroimine solution markedly inhibited the ACh-evoked current (IACh), as here exemplified with the (±)-**4** racemate (Figure 2a *right*, tracing 2). When the α1_2_βγδ nAChR was in the desensitized state, the washout of the bound spiroimine with ACh triggered the immediate reopening of the receptor channel and full recovery of the nicotinic current up to the amplitude level of the control desensitized state, a feature denoting fast reversibility of channel blockade by the spiroimine (Figure 2a *right*, tracing 3). 

Concentration–response curves recorded on the α7 nAChR also showed a full antagonistic effect at high spiroimine concentration, with IC_50_ values in the *medium* µM range, and slightly higher, albeit not statistically different, for the (+)-**4 R** enantiomer compared to its two relatives (Figure 2b *left*; Table 1). Unexpectedly, negative cooperativity was observed, higher for the (−)-**4 S** enantiomer than the (+)-**4 R** enantiomer, and averaged for the (±)-**4** racemate, suggesting the contribution of an allosteric component to α7 antagonism by the spiroimines along with stereospecific modes of binding. Here again, recovery of the ACh-evoked current following the washout of bound spiroimine was fast and complete, as here exemplified with enantiomer (−)-**4 S** (Figure 2b *right*, tracing 3). 

Similar antagonistic potency trends were again obtained for the α4β2 nAChR, with IC_50_ values in the *medium-to-high* µM range, but now significantly higher, by ca. one order of magnitude, for both the (±)-**4** racemate and (+)-**4 R** enantiomer compared to their (−)-**4 S** relative (Figure 2c *left*; Table 1). Here again, negative cooperativity was observed, higher for the more potent (−)-**4 S** enantiomer than for the (+)-**4 R** enantiomer and (±)-**4** racemate. Recovery of the ACh-evoked current was fast and complete, as here exemplified with enantiomer (+)-**4 R** (Figure 2c *right*, tracing 3; Figure 3a) and with the (±)-**4** racemate using a slightly different protocol (Figure 3b).

Overall, these data point to similar antagonistic potencies of the three spiroimines for the muscle-type α1_2_βγδ nAChR and a slightly higher potency of the (−)-**4 S** enantiomer for the neuronal α7 and α4β2 nAChR subtypes compared to its two relatives, with a one order of magnitude stereospecificity for α4β2. However, they also point to a higher selectivity, by 13- and 25-fold, of the (+)-**4 R** enantiomer for the α1_2_βγδ nAChR compared with the α7 and α4β2 subtypes. The IC_50_ values recorded for the (±)-**4** racemate were overall consistent with the presence of the two enantiomers in a 1:1 ratio (see *Experimental Procedures*). 

These data also support the role of the spiroimine moiety in the cyclic imine toxins as a main pharmacophore, as previously approached using 6,6-spiroimine analogs of GYM [74], whereas a synthetic tetrahydrofuran fragment mimicking the C10–C20 skeleton of GYM was found not to bind the α1_2_βγδ nAChR [73]. Hence, developing spiroimine-like bioactive molecules may be a promising strategy for designing new effectors targeting disease-associated nAChR subtypes. In turn, the weak antagonistic potency of the spiroimines, lower than that of GYM by two (α1_2_βγδ) to three (α7) to four (α4β2) orders of magnitude [71,77], likely proceeds from a much shorter residence time (i.e., greater k_off_ value) within the orthosteric binding site of the nAChRs due to the limited number of possible interactions with the subunit interface enabled by their small molecular size. Competition binding studies on nAChRs expressed at the surface of mammalian cells were not performed; however, these parameters were explored through binding studies on A- and L-AChBP and the structural analysis of spiroimine-A-AChBP complexes (see below). 

### 2.3. Binding Characteristics of the Spiroimine Enantiomers toward A- and L-AChBPs

The sequence differences and distinctive binding affinities of A- and L-AChBP for nicotinic ligands render them useful templates for approaching ligand binding to various nAChR subtypes. These differences are exemplified by the A-AChBP lower affinity for ACh, but higher affinity for some α7-specific peptidic antagonists, a feature that renders it more “α7-like” than L-AChBP [63,64,69,70].

Equilibrium dissociation constants were determined either from ratios (K_d/(koff/kon)_) of rate constants determined using multiple kinetic means (k_on_, k_off_, k_off/GAL_) or directly from stopped-flow measurements (K_d/SFeq_, K_d/SPAeq_) (see *Experimental Procedures*). Overall, interaction of the spiroimines with each AChBP was found to be clearly stereospecific (Table 2). Specificity was even more pronounced with L-AChBP with a one-to-two orders of magnitude difference between the equilibrium constants (K_d_ values) of the (+)-**4 R** and (−)-**4 S** enantiomers. This difference most likely arises from the slower association (lower k_on_ values) and faster dissociation (higher k_off_ value) of enantiomer (+)-**4 R**, as also suggested by the kinetic values obtained for racemic (±)-**4** and reflected in the fraction of the complex too low for reliable detection under the experimental conditions of the stopped-flow measurement. Enantiomer (+)-**4 R** was a tighter binder to A-AChBP, albeit with a modest stereospecificity of up to two-fold arising largely from slower complex dissociation (lower k_off_ values). In turn, enantiomer (−)-**4 S** bound tighter to L-AChBP with K_d_ values in the 10^−7^ M range, resulting in the tightest complex of all combinations, while enantiomer (+)-**4 R** bound with K_d_ values in the 10^−5^ M range, not permitting the separation of rate constants for complex formation and dissociation under pre-equilibrium conditions.

The interaction constant values determined by the different experimental approaches generally agreed well (Table 2). Both the k_off_ and k_off/GAL_ values and the K_d/(koff/kon)_, K_d/SFeq_ and K_d/SPAeq_ values determined in differently designed assays were comparable. Only the K_d/SPAeq_ value determined for enantiomer (−)-**4 S** binding to L-AChBP was ca. one order of magnitude higher than the K_d/(koff/kon)_ and K_d/SFeq_ values determined in two different stopped-flow based assays (performed under pre-equilibrium and equilibrium conditions of spiroimine interaction to AChBP). Hence, this divergent value may have to be considered with caution, although in terms of the associated ΔΔG value, the difference is not significant. Quite surprisingly however, the higher enantioselectivity for L-AChBP was also reflected in the biphasic curve obtained upon stop-flow recording of the binding rates (K_d/SFeq_ values). The lower enantioselectivity (i.e., comparable affinities) of enantiomers (+)-**4 R** and (−)-**4 S** for A-AChBP (K_d_ values in the 2.2–5.4 µM range) compared to L-AChBP (12–0.32 µM range) justified the use of A-AChBP for the structural analyses.

### 2.4. Overall View of the Crystalline Spiroimine-AChBP Complexes

The structures of A-AChBP bound with spiroimines (+)-**4 R** and (−)-**4 S** were solved in the 1.85–2.00 Å resolution range from preformed stoichiometric complexes (Appendix A; Figure 4). Both showed the same tight homopentameric ring assembly of subunits as found in all previous AChBP structures (for reviews, see [69,70]). The ligand binding pocket encompasses a nest of five electron-rich aromatic side chains provided by residues Tyr93, Trp147, Tyr188 and Tyr195 on the principal (+) face of the subunit interface and residue Tyr55 on the complementary (−) face. This pocket is partially sheltered from the solvent by loop C, which is found at the outer perimeter of the pentamer and harbors at its tip a disulfide bridge linking the vicinal Cys190 and Cys191 residues, linked into a Cys-*trans*-Cys disulfide bridge (*aka* oxidized cysteinyl-cysteinyl ring), a signature determinant for nAChR α subunits.

Both structures showed very similar positions of the bound spiroimines at all five subunit interfaces within a pentamer (see. r.m.s.d. values in *Experimental Procedures*). Despite the moderate binding affinities (Table 2), the well-defined electron densities revealed full ligand occupancy at all five binding sites (Figure 4) due to the high protein concentration (ca. 100-fold the K_d_ values) and slight molar excess of spiroimine over A-AChBP used for complex formation and crystallization (see *Experimental procedures*).

### 2.5. Detailed Description of the Crystalline Spiroimine-AChBP Complexes

In the A-AChBP complex with enantiomer (+)-**4 R**, the compact 3-ring skeleton of the spiroimine comfortably accommodates the core agonist signature motif central to the binding pocket (Figure 4 and Figure 5) and ideally positions the 6-membered cyclic imine ring nearly parallel to the indole ring of Trp147 in loop B (π-π stacking), and within H bond distances (2.6–2.7 Å) of the Trp147 carbonyl oxygen, with a typical *Ĥ* angle value of ~125° between the hydrogen bond donor, acceptor, and acceptor antecedent [78] (Appendix A), as previously observed for the parental macrocyclic imine toxins (Figure 6). Such conservation of a *protonated* imine nitrogen (see discussion in [74] and reference 18 in it, along with the crystallization conditions in the *Experimental procedures*) tethers the spiroimine core centered within the binding pocket and greatly contributes to the binding affinity. The dioxolane ring makes nearly edge-to-face stacking interactions with Tyr188 and Tyr195 (loop C) from the (+) face, while the cyclohexane ring makes nearly face-to-face stacking interactions with Tyr93 from the (+) face and Tyr55 (loop D) from the (−) face (Figure 4 and Figure A1). However, spiroimine binding involves virtually no interaction with residues from loop F at the complementary (−) face of the subunit interface (Figure 4, Figure 5 and Figure A1). This observation is consistent with the limited contribution of loop F to the binding of SPX and GYM *versus* its significant contribution for laterally accommodating the bridged ketal group specific to the pinnatoxins [71,72] (Figure 6).

In the A-AChBP complex with enantiomer (−)-**4 S**, the cyclic imine ring, nearly perpendicular to the indole ring of Trp147 (T-shaped interaction), only partially overlaps that of the (+)-**4 R** enantiomer, with long-range H bond distances (2.8–3.5 Å) to the carbonyl oxygen of Trp147 and an *Ĥ* angle value of ~142° (Figure 4 and Figure 5; Appendix A). As a result, the position of the cyclohexane ring is displaced toward the tip of loop C, and the dioxolane ring projects in the opposite direction at the entrance of the binding pocket, thus making limited interactions with the A-AChBP residues. Such a drastic shift in the position of the 3-ring system for the two enantiomers is associated with differences in the loop-C position (Figure 4, Figure 5 and Figure 6, and see below) and may account for the slightly (ca. 2-fold) higher affinity of enantiomer (+)-**4 R** for the two AChBPs compared to enantiomer (−)-**4 S** (Table 2). In turn, the one order of magnitude higher affinity of enantiomer (−)-**4 S** for L-AChBP compared to A-AChBP (Table 2; Figure 6b and Figure A1) may correlate with the presence of substitutions Met114 (for Ile118) and Trp53 (for Tyr55) in the vicinity of the cyclic imine and cyclohexane rings. This is consistent with the ca. 5-fold higher affinity of enantiomer (+)-**4 R** compared with the (−)-**4 S** enantiomer for the α7 and α4β2 nAChRs (Table 1; Figure A1). Here again, there is almost no interaction between the bound spiroimine and loop-F residues (Figure 4, Figure 5, and Figure A1).

Co-crystallization of A-AChBP with the (±)-**4** racemate unambiguously led to a bound (+)-**4 R** enantiomer at all five subunit interfaces in a virtually identical position to that seen in the (+)-**4 R** complex, with only a minor trace (i.e., occupancy below ~10%) of a bound (−)-**4 S** (Figure 4d). Per se, this observation would suggest a much lower dissociation rate for enantiomer (+)-**4 R** relative to its (−)-**4 S** congener than the one recorded through the binding studies (Table 2). Whether this is due to the more acidic and hydrophobic composition of the crystallization liquor used for the (±)-**4** complex compared with the other two complexes (see Appendix E) or to conformational remodeling of the complex during crystal nucleation and growth [79] is unknown.

In all three complexes, the loop-C position shows a high degree of variability with respect to the different orientation of the 3-ring system in each enantiomer associated with slightly different orientations of the bound enantiomers at each binding interface (Figure 4c). In the (+)-**4 R** complex, loop C adopts a *closed*, agonist-bound position in one subunit along with a range of positions *intermediate* between those found in apo (2BYN) and agonist-bound A-AChBP (2BYQ, epibatidine) in the other four subunits. In the (±)-**4** and (−)-**4 S** complexes, loop C clusters around *intermediate* positions in all five subunits. Moreover, at one interface of the (−)-**4 S** complex, the tip of loop C (Tyr188-Glu193) adopts two alternate conformations with up to 3 Å distance at position Cys190, an observation emphasizing the high conformational dynamics of loop C, even in the crystalline state. The average loop-C conformation in the spiroimine-AChBP complexes contrasts with the wide-open, “antagonist” position observed in the complexes with the parental, bulky macrocyclic spirolides/phycotoxins [71,72] or other large antagonists (Figure 6). However, the loop-C “intermediate” positions associated with its loose interactions with the spiroimines resemble more the position observed in HEPES-bound or apo AChBPs [63,67,80] than that in complexes with small agonists [65,67,68] (Figure 6). A reverse situation was reported for 2-aminopyrimidine agonists inducing an unusually large opening of loop C [81]. This comparative analysis further reflects the antagonist properties of these small spiroimines and illustrates, *here within a single AChBP pentamer*, how loop C can behave as a highly flexible sensor to adapt its configuration to the chemical features and position of the ligand within the binding pocket, as previously documented for many structurally-unrelated nicotinic ligands.

To explore a possible correlation between the spiroimine binding poses and interactions at the A-AChBP subunit interfaces and their binding affinities in each complex, from the structure coordinates, we calculated the DrugScore eXtended (DSX) scores, which combine distance-dependent atom–atom potentials, torsion angle potentials, and solvent-accessible surface-dependent potentials [82] (Table 3).

The smaller mean score for the interaction of enantiomer (+)-**4 R** at all five A-AChBP subunit interfaces compared to that for enantiomer (−)-**4 S** suggests tighter binding to A-AChBP, independently of the loop-C conformation. The ratio of the (+)-**4 R**/(−)-**4 S** mean scores, equal to 1.22, also indicates computational prediction from the structures for a tighter binding of enantiomer (+)-**4 R**, which should be reflected by a ca. one order of magnitude difference in the Kd values for the two complexes [82]. The fact that our experimentally determined values do not differ that much from each other (Table 2) may arise either from differences in the “experimental” conditions or from five interfaces per complex making too small a statistical sample, or from the limited significance of comparing differences in affinities in the micromolar range, or, here again, from time-dependent conformational remodeling of the complexes in the crystal state but not during the binding experiments in solution [79]. Overall, the DX scoring approach appears consistent with the prevailing presence of enantiomer (+)-**4 R** in the structure of the complex prepared from the (±)-**4** racemate (Figure 4). 

### 2.6. Structural Comparisons

Certain features of the A-AChBP complexes with the (+)-**4 R** and (−)-**4 S** enantiomers closely resemble those found in the complexes with the bulkier macrocyclic imine compounds GYM and SPX (see r.m.s.d. values in *Experimental Procedures*), with in particular, the positions of the cyclic imine and cyclohexene rings (Figure 6). In fact, the orientation of the cyclic imine ring in the (−)-**4 S** complex virtually overlaps that of GYM. Structural overlay of these two complexes shows that the lack of the bis-spiroacetal/tetrahydrofuran and butyrolactone ring systems of GYM in the apical and membrane directions of the subunit interface, respectively, is partially compensated by the new dioxolane ring in the spiroimines that abuts against the Tyr188 and Tyr195 phenol rings in loop C. The loop-C conformational, large-amplitude motions observed across the five binding interfaces in the (+)-**4 R** and (−)-**4 S** complexes, coupled with alternate conformations of its tip (a behavior neither observed in complexes with macrocyclic phycotoxins nor in those with small agonists), appear to be a signature of the minimal spiroimine core motif. This comparison also suggests substituent modification at key positions around the 3-ring framework of spiroimines, notably in the cyclohexane ring, to confer greater binding selectivity and specificity on them. For example, a polar group attached to the cyclohexene ring would be expected to target residues in or near loop F at the complementary face of the subunit interface.

Comparisons with humanized A-AChBP-α7-chimeras in the apo conformation [80] and the extracellular LBDs of the α4β2, α7, and α1_2_βγδ nAChRs [59,60,83] showed a conserved aromatic nest with the exception of a Trp (in α4 and α7) or Arg (α1) residue in place of A-AChBP Tyr55 at the (−) face of the interface, and the positions of Tyr188 and Tyr195 within loop C at the (+) face (Figure 6 and Figure A1; Appendix A). Moreover, conservation of the “interacting” residues at the αγ and αδ interfaces of the α1_2_βγδ nAChR is consistent with the Hill slope of ~1 observed in the functional assays (Figure 2, Figure 6 and Figure A1; Appendix A). Concerning the α3β2 nAChR, which was not included in the present study, a similar binding mode and antagonistic potency as for the α4β2 subtype can be predicted due to their high sequence homology in the spiroimine binding site (see Figure S1 in [70]).

## 3. Experimental Procedures

### 3.1. Chemical Synthesis, Separation, and Enantiomeric Characterization of the Spiroimines

Racemic spiroimine was synthesized from diketone **1** by alkylation with 1-azido-3-iodopropane in the presence of anhydrous cesium carbonate to afford the azido-ß-keto ketone (±)-**2** in 53% yield (Figure 1; Appendix C). The cyclohexanone function of (±)-**2** was then protected with ethylene glycol to generate the corresponding dioxolane (±)-**3** in 31% yield. Then, the azide function of (±)-**3** was reduced with triphenylphosphine, followed by cyclization under basic conditions to generate spiroimine (±)-**4** in 83% yield. Each compound was characterized by ^1^H and ^13^C NMR spectroscopy in CDCl_3_ (Appendix A). The two enantiomers present in a 1:1 ratio in (±)-**4** were then separated by semi-preparative chiral HPLC in Hept/EtOH 80:20 (*v*/*v*) to afford (+)-**4** (*t* = 4.54 min, $\alpha_{D}^{24}$ = +144.5) (c = 0.25, CHCl_3_) and (−)-**4** (*t* = 6.21 min, $\alpha_{D}^{24}$ = −163.5) (c = 0.25, CHCl_3_) with a purity greater than 99% (Appendix A). The racemate and pure enantiomers were lyophilized and stored at −20 °C. Stock solutions at 10 mM (racemate) and 20 mM (enantiomers) in MeOH prepared from the lyophilized compounds were stored as aliquots at −80 °C.

The synthesis of enantiomer (−)-**4** from ketone **5** (a mixture of (+)-**5 S** and (−)-**5 R** in a 90:10 molar ratio, prepared according to [76]), was adapted from [74]. In brief, Ru-catalyzed cross-metathesis with vinyl pinacol boronate (step 1), followed by oxidation, then reduction (step 2), afforded the primary alcohol (not shown). The latter was converted into the azide (step 3), which was reduced with triphenylphosphine (step 4) to give spiroimine (−)-**4** ($\alpha_{D}^{24}$ = −47.3 (c = 0.25, CHCl_3_) (Figure 2), of absolute configuration (S) at the quaternary carbon, as indicated by the negative sign of the optical rotation. 

### 3.2. Analysis of ACh-Evoked Currents

The sources for live animals and biological materials, the procedures for nAChR microtransplantation and expression in oocytes, and the procedures for manual (α1_2_βγδ nAChR) and automated (α7 and α4β2 nAChRs) TEVC recording are detailed in Appendix D. In brief, our TEVC protocol comprised 2 to 3 pulses of ACh at a concentration equivalent to its EC_50_ value, such as: 15 s at 25 µM for α1_2_βγδ nAChR, 5 s at 100 µM for α7 nAChR, 15 or 30 s at 150 µM for α4β2 nAChR (perfusion flow, 8 mL/min). Then, the oocyte was perfused for 45 s with the tested spiroimine at a given concentration, immediately followed by the co-application of ACh and the spiroimine at the same concentrations and times as indicated above. The oocyte was washed with Ringer’s solution, and then recovery of the ACh-evoked current was recorded.

The concentration–inhibition curves were analyzed as previously detailed [71,72] using the equation I = I_max_ [L]^nH^/(EC_50_^nH^ + [L]^nH^) (Equation (1)), where I is the measured agonist-evoked current, [L] is the agonist concentration, EC_50_ is the agonist concentration that evoked half the maximal current (I^max^), and nH is the Hill coefficient. For antagonist inhibition, current (I) values were normalized to the I^max^ value recorded from the same oocyte to yield the fractional (%) response data. IC_50_ values were established from the concentration–response curves by fitting to the equation F = 1/[1 + ([X]/IC_50_)^nH^] (Equation (2)), where F is the fractional response obtained in the presence of the inhibitor at concentration [X], and IC_50_ is the inhibitor concentration reducing the ACh-evoked amplitude by half. Data analysis was performed without constraints as Log[spiroimine] versus response (three parameters), with a Hill slope ~1. Statistical significance of differences between the control and test values was assessed using either the two-tailed Student’s “*t*” test or the Kolmogorov–Smirnov two-sample test and *p* < 0.05.

### 3.3. Ligand Binding to the AChBPs

The procedures for stable expression and purification of A- and L-AChBPs are described in Appendix E. Rate constants for association (k_on_) and dissociation (k_off_) were determined by multiple kinetic means, largely as previously described [64,67,71,72]. Measurement of k_on_ and k_off_ values entailed the direct admixture of reactants and monitoring the quenching of AChBP native Trp fluorescence. Individual rate constants were determined from linear regression of the experimental mono-exponential decay of Trp fluorescence intensity (k_obs_) using the linear “approach-to-equilibrium” relationship k_obs_ = k_on_[spiroimine] + k_off_ (Equation (3)). Additional measurements of k_off_ (k_off/GAL_) employed stopped-flow measurement of the rate of occupation of free AChBP binding sites by a competing ligand (gallamine, used in large excess relative to its K_d_) to form a non-quenching gallamine-AChBP complex. Equilibrium dissociation constants were determined either from the ratios of rate constants (K_d/(koff/kon)_), or from stopped-flow measurements of the rates of gallamine or epibatidine binding to pre-equilibrated spiroimine-AChBP complexes formed using increasing spiroimine concentrations (K_d/SFeq_), or by the scintillation proximity assay (SPA) where spiroimine competition against the binding of [^3^H]epibatidine (to L-AChBP) or [^3^H]methyllycaconitine (to A-AChBP) was monitored at equilibrium (K_d/SPAeq_). In brief, in the SPA assay, to titrate those binding sites made available upon toxin dissociation, each equilibrated spiroimine-AChBP complex, at 250 pM in binding sites and a slight molar excess of toxin, was mixed with the [^3^H]ligand at a concentration well above its K_d_ value. The time course of [^3^H]ligand binding was monitored over several hours. All experiments were performed in triplicate, in which individual data differed by less than 20%.

### 3.4. Structure Determination and Refinement

The procedures for the formation and crystallization of the spiroimine-A-AChBP complexes and for data collection are described in Appendix E. The structure of the A-AChBP complex with spiroimine (+)-**4 R** was solved by molecular replacement with PHASER [84] using the apo A-AChBP structure (PDB: 2BYN [67]) as a search model, and that of the complex with spiroimine (−)-**4 S** by difference Fourier synthesis with REFMAC [85]. The models of the spiroimine-A-AChBP complexes were improved by manual adjustment with COOT [86] and refined with REFMAC including TLS refinement with each subunit defining a TLS group. In each case, a random set of reflections was set aside for cross-validation purposes. The molecular structures of spiroimines (+)-**4 R** and (−)-**4 S** and the associated library files containing the stereochemical and parametric data were generated with SKETCHER [87]. Ligands were fitted into unbiased Fo–Fc difference electron density maps calculated after 10 cycles of rigid-body refinement. Automated solvent building used COOT. The A-AChBP-spiroimine interfaces were analyzed with the PISA server [88]. Data collection and refinement statistics are reported in Appendix A.

The final structures comprise A-AChBP residues His1-Arg207/Arg208 for each subunit in the single homopentamer present in the asymmetric unit, and a fully resolved spiroimine in each of the five binding sites. For each complex, the N-terminal FLAG sequence could be fully resolved in one of the five subunits, and a tetrasaccharide moiety linked to Asn74 was visible in another subunit. High temperature factors and weak electron densities were associated with the other four FLAG sequences and with the five surface loops Asn15–Met19. Five chlorine ions in the (+)-**4 R** complex and one isopropanol molecule and six chlorine ions in the (−)-**4 S** complex (all arising from the crystallization liquor or the purification buffer) could also be modeled. Quality of the models was validated using COOT and MOLPROBITY [89], with ~97% of residues in favored regions of the Ramachandran plot and no outliers. Data collection and refinement statistics are reported in Appendix A. 

### 3.5. Structural Analyses and Comparisons

Comparison of the spiroimine-A-AChBP complexes with other AChBP structures included those of A-AChBP in the apo form (PDB: 2BYN [67]) and bound with phycotoxins SPX and GYM (2WZY and 2X00 [71]), pinnatoxins PnTxA and PnTxG (4XHE and 4XK9 [72]), and the small agonist representative, EPI (2BYQ [67]); of L-AChBP in the “apo” (HEPES-bound) form (1I9B [63]); and of humanized A-AChBP-α7-chimeras (mutants I and II; 3T4M and 3SIO [80]. The average root-mean-square deviation (r.m.s.d) between A-AChBP subunits bound with spiroimine (+)-**4 R** with *closed* and *intermediate* states of loop C is 0.6 Å; between the five A-AChBP subunits bound with spiroimine (−)-**4 S** with *intermediate* states of loop C, it is 0.27 Å; between the A-AChBP subunits bound to spiroimine (+)-**4 R** with a *closed* loop C versus bound to spiroimine (−)-**4 S** with an *intermediate* state of loop C, it is 0.43 Å; and between the nine A-AChBP subunits with *intermediate* states of loop C, it is in the 0.27–0.43 Å range (all for 204–210 Cα atoms).

Comparisons with structures of full-length nAChRs included the human α7 bound to EPI (cryo-EM structure 7KOQ [59]), the human α4β2 in the apo form (X-ray structure 5KXI [60]), and the *Torpedo* α1_2_βγδ receptor in the apo form (cryo-EM structure 7SMM [83]). The average r.m.s.d. between A-AChBP subunits bound with spiroimine (+)-**4 R** with a *closed* loop C or with spiroimine (−)-**4 S** with an *intermediate* loop C and the extracellular ligand-binding domains of the above-mentioned nAChRs is in the 1.3–1.7 Å range for 181–194 Cα atoms.

### 3.6. Figures

Figure 1 was generated with Marvin (ChemAxon); Figure 1 and Figure 2 and those in Appendix C with ChemDraw; Figure 2 and Figure 3 with GraphPad Prism 9.0 (GraphPad Software, San Diego, CA, USA) and pCLAMP 9.0 (Molecular Devices, LLC., San Jose, CA, USA); Figure 4, Figure 5 and Figure 6 with PyMOL [90]; Figure A1 with ESPript [91]; Appendix A with TopSpin; and Appendix A with MassLynx.

## 4. Concluding Remarks

These data emphasize the intrinsic capacity of the spiroimine enantiomers synthesized in this study to block selected nAChR subtypes in a competitive manner, highlight the major contribution of the spiroimine core of macrocyclic imine toxins to binding within the nAChR aromatic nest (*aka* “agonist-competitive antagonist site” [92]), and confirm the need for extended interaction networks as established by the macrocyclic toxins to define high affinities and the variable levels of subtype specificity dictated by their capacity to extend in all directions of the subunit interface. Hence, this study identifies a minimal set of functional pharmacophores and binding determinants (Appendix A) as templates for designing second-generation spiroimine-like bioactive molecules with full antagonistic properties targeting disease-associated nAChR subtypes. From that perspective, enantiomer (+)-**4 R**, with its higher specificity for the α1_2_βγδ nAChR compared with the α7 and α4β2 subtypes, and its tighter binding to key residue Trp147 in the A-AChBP binding pocket compared with enantiomer (−)-**4 S**, appears as a suitable starting point in a context of muscle-linked diseases.

These data also contribute to challenge a conclusion built over the years that small molecules containing a quaternary nitrogen and strictly occupying the agonist binding pocket at the subunit interface, without extending substituents outside of this core, cannot act as nicotinic antagonists. Initial insights against this “dogma” include the small cyclic, disulfide-containing marine molecule, nereistoxin (*aka N*,*N*-dimethyl-1,2-dithiolan-4-amine), shown to act as a nicotinic antagonist with sub-millimolar IC50 values [93]. Availability of a crystal structure of a nereistoxin-AChBP complex would be of interest to complement the current study.

## Data Availability

The data presented in this study are included in the main text, Appendix B, Appendix B, Appendix B and Appendix B, and the Appendix A. The atomic coordinates and structure factors of the A-AChBP complexes with enantiopure spiroimines (+)-**4 R** and (−)-**4 S** and the (±)-**4** racemate have been deposited at the RCSB-PDB (www.rcsb.org) with accession codes 8Q1M, 8QTL, and 8QX2, respectively.

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
