# Peer review of "The Cyclic Imine Core Common to the Marine Macrocyclic Toxins Is Sufficient to Dictate Nicotinic Acetylcholine Receptor Antagonism"

_marinedrugs, 2024, doi:10.3390/md22040149_

Round 1
Reviewer 1 Report
Comments and Suggestions for Authors
The manuscript entitled “The Cyclic Imine Core Common to the Marine Macrocyclic Toxins is Sufficient to Dictate Nicotinic Acetylcholine Receptor Antagonism” reports the results of a multidisciplinary investigation (organic synthesis, electrophysiology, kinetic, thermodynamic measurements and crystallographic studies) of original chiral derivatives mimicking the pharmacophore of phycotoxins targeting nAChR and their properties/characteristics. The manuscript is very well written, gathers international recognized experts of the field and provides a lot of relevant informations for such investigations: chemical synthesis and characterization of the compounds, electrophysiology measurements of their activities as racemate and enantiomers on different nAChRs subtypes; binding studies with A- and L-AChBPs and structural characterization of co-crystals of spiroimine-AChBP complexes. The work provides original detailed informations that might help in the design of new antagonists targeting disease-associated nAChR subtypes.
Despite these remarks, several points should be considered and further developed and or resolved by the authors.
1. First, the difference in terms of targets used in the various parts of the study, even if justified in some way by the authors, raises question. Thus, the electrophysiology measurements have been carried on three nAChRs subtypes whereas the binding and structural studies have been realized using A- and L-AChBPs. For such subtle investigations (reduced parts of the toxins, with chiral features), it deserves more discussion. In my point of view, the sentence at the end of the first paragraph of page 3 …: “However, despite the remarkable technological and experimental progress, using full- length receptors remains both time and cost consuming, and risky in terms of success” is not sufficient. Even if some elements are given in the following paragraph with respect to the homology between AChBP and the LBD of nAChR, more details should be given. Despite the homology shown in Figure A1, from a functional point a view, there are major differences between AChBP and nAChR that should be commented in the context of this study.
2. In relation to this, I could not find any justification of the use of two AChBP (A- and L-) for the study. Could the authors justify this ?
3. Page 12 : in paragraph 2.5, the authors provide a detailed decription of the cristallographic spiroimine-AChBP complexes. In this paragraph, they consider the 6-membered cyclic imine of the spiroimine ring protonated. Is there a rationale behind this ? The pka of imines is around 7 and therefore the proportion of protonated and neutral forms should be nearly equal ? Could the authors provide arguments for the consideration of the protonated form?
4. Page 14 : Table 3. the units of the DSX scores ( kcal/mol or kJ/mole) should be specified
5. In complement to the discussion related to the scores, a table comparing geometric parameters of key interactions of the various enantiomers with the binding site residues of A-AChBP (e.g. between the imine nitrogen and the Trp147 carbonyle oxygen) should be useful for the reader.
For the reasons mentioned above, the manuscript should be accepted with minor changes, corresponding to the modifications required to answer the points raised above.

Author Response
We thank this reviewer for the positive evaluation and constructive comments and suggestions. Here, our replies appear in green below each of them. The associated modifications in the text, along with others of our own aimed at improving the manuscript further (including 7 references), are highlighted in yellow in the revised manuscript.
1. First, the difference in terms of targets used in the various parts of the study, even if justified in some way by the authors, raises question. Thus, the electrophysiology measurements have been carried on three nAChRs subtypes whereas the binding and structural studies have been realized using A- and L-AChBPs. For such subtle investigations (reduced parts of the toxins, with chiral features), it deserves more discussion. In my point of view, the sentence at the end of the first paragraph of page 3: “However, despite the remarkable technological and experimental progress, using full- length receptors remains both time and cost consuming, and risky in terms of success” is not sufficient. Even if some elements are given in the following paragraph with respect to the homology between AChBP and the LBD of nAChR, more details should be given. Despite the homology shown in Figure A1, from a functional point a view, there are major differences between AChBP and nAChR that should be commented in the context of this study.
The functional differences between AChBP and nAChR are now mentioned in the Introduction (page 3, second paragraph).
2. In relation to this, I could not find any justification of the use of two AChBP (A- and L-) for the study. Could the authors justify this?
The use of two AChBPs for our binding studies is now justified at the beginning of section 2.3.
3. Page 12 : in paragraph 2.5, the authors provide a detailed decription of the cristallographic spiroimine-AChBP complexes. In this paragraph, they consider the 6-membered cyclic imine of the spiroimine ring protonated. Is there a rationale behind this? The pka of imines is around 7 and therefore the proportion of protonated and neutral forms should be nearly equal? Could the authors provide arguments for the consideration of the protonated form?
This point had been discussed previously for two other spiroimine compounds (ref [75] Duroure et al. 2011, middle part of the discussion), and co-crystallization of our complexes was achieved at pH ≤ 7.5 (see Experimental procedures). To make it clear we introduced a comment in our revised manuscript (page 12, paragraph 2.5.).
4. Page 14 : Table 3. the units of the DSX scores (kcal/mol or kJ/mole) should be specified.
Neither the computer application that we used to determine DSX scores nor the Neudert & Klebe article (ref. 81) where the DSX concept was described, offer units for the scores. Direct exchanges with program developers and with Prof. Klebe led us agree on that DSX scores represent relative interaction energies given in kJ/mol, of small ligands with a receptor. Therefore we added units "(kJ/mol)" in the table header, along with an explanatory sentence in the Table 3 footnote (page 14).
5. In complement to the discussion related to the scores, a table comparing geometric parameters of key interactions of the various enantiomers with the binding site residues of A-AChBP (e.g. between the imine nitrogen and the Trp147 carbonyl oxygen) should be useful for the reader.
A new Table S2 entitled “Geometric parameters of key interactions between the (+)-4 R and (–)-4 S enantiomers and side chains in the A-AChBP binding pocket” has been added as Supplementary Material and is now referred to in Section 2.5., pages 12-13.
For the reasons mentioned above, the manuscript should be accepted with minor changes, corresponding to the modifications required to answer the points raised above.
We hope the revised version of our manuscript now meets this reviewer’s expectations.
Reviewer 2 Report
Comments and Suggestions for Authors
This is an excellent article that reports an investigation of the importance of the spiroimine moiety for the interaction of spiroimine marine toxins with three different nicotinic ACh receptors and with AChBP. It demonstrates the primary importance of the cyclic imine moiety for binding to the ACh recognition site. The ability of such small molecules to act as antagonists indicates that large ligand size is not required for antagonism. The preferential binding of one of the enantiomers is demonstrated and crystallographic data offers an explanation for this.
Author Response
We thank this reviewer for the fully positive evaluation and kind words.
Reviewer 3 Report
Comments and Suggestions for Authors
The manuscript describes the attempt to use core macrocyclic imine phycotoxins as a base for creation of selective nAChR antagonists. Big bulk of data presented, but there are some remarks about analysis and interpretation of results.
Major concerns
1. Introduction. Illogical consequence of paragraphs, should be re-written. Should be: toxins+ their action on nAchR, then some important facts about nAChR (if necessary).
2. Results.
2.2 Fig 2, 3 and Table 1. Something probably wrong in the fitting of (–)-4 S for α7 and α4β2. I think the authors fixed the maximal and minimal response at 100% and 0 during the fitting procedure, that is not fully correct. Authors should try to analyze the data correctly, and that could give them additional material for discussion.
Additionally, left panel Fig 2 and Fig 3 marked by numbers, but it was much better to write the actual information, there is plenty of space.
Authors decided to make results and discussion, but I see results and do not see discussion. Why different methods of activity evaluation give different results? Which compound is more suitable for further design of selective antagonists?
Conclusion remarks are enthusiastic, but they should have some data in the manuscript to support it.
"Hence, this study identifies a minimal set of functional pharmacophores and binding determinants as templates for designing second-generation spiroimine-like bioactive molecules with full antagonistic properties targeting disease-associated nAChR subtypes." - There was no significant difference between compounds in the most of the tests. In some tests, you have got opposite results. So, what is the minimal set of functional pharmacophores and binding determinants?
I think the manuscript contains interesting information. But should be carefully re-written to improve the quality of results presentation, discussion and conclusions.
Comments on the Quality of English LanguageMinor editing is needed to improve English.
Author Response
We thank this reviewer for the positive evaluation and constructive comments and suggestions. Here, our replies appear in green below each of them. The associated modifications in the text, along with others of our own aimed at improving the manuscript further (including 7 references), are highlighted in yellow in the revised manuscript.
Major concerns
1. Introduction. Illogical consequence of paragraphs, should be re-written. Should be: toxins+ their action on nAchR, then some important facts about nAChR (if necessary).
We carefully considered the reviewer’s suggestion. However, we chose not to rewrite the introduction (which the other two reviewers seemed to have found appropriate) but rather to introduce a few sentences making a better link between the “toxins” and “nAChRs” paragraphs and citing suitable reviews (page 2, beginning of bottom paragraph).
2. Results.
2.2 Fig 2, 3 and Table 1. Something probably wrong in the fitting of (–)-4 S for α7 and α4β2. I think the authors fixed the maximal and minimal response at 100% and 0 during the fitting procedure, that is not fully correct. Authors should try to analyze the data correctly, and that could give them additional material for discussion.
We checked the fitting of all curves in Fig 2 and corrected those that were not accurate (new Fig. 2). We modified the associated values and added the Hill slopes in Table 1. And we modified the associated data description (page 7, below table 1; page 8, below Fig 3).
Additionally, left panel Fig 2 and Fig 3 marked by numbers, but it was much better to write the actual information, there is plenty of space.
We added the requested information on Figs. 2 & 3 (even though we feel it cluttered them).
Authors decided to make results and discussion, but I see results and do not see discussion.
We developed the discussion at several places within the Results and Discussion section.
Why different methods of activity evaluation give different results?
We assume that this comment is related to Table 2 and that “activity” should read “binding”. If so however, the results obtained from our differently designed assays are not “different”. Only one of the Kd values determined for enantiomer (–)-4 S binding to L-AChBP is somehow higher than the other two values; yet in terms of ΔΔG values the difference is negligible. This point is now discussed (page 9, bottom paragraph).
Which compound is more suitable for further design of selective antagonists?
A sentence pointing to enantiomer (+)-4 R as a suitable starting point in a context of muscle-linked diseases has been added in the conclusion (Concluding remarks, page 18).
Conclusion remarks are enthusiastic, but they should have some data in the manuscript to support it. “Hence, this study identifies a minimal set of functional pharmacophores and binding determinants as templates for designing second-generation spiroimine-like bioactive molecules with full antagonistic properties targeting disease-associated nAChR subtypes.” – There was no significant difference between compounds in the most of the tests. In some tests, you have got opposite results. So, what is the minimal set of functional pharmacophores and binding determinants?
The new Table S2 and the new sentence added in the conclusion should now clearly define the point.
I think the manuscript contains interesting information. But should be carefully re-written to improve the quality of results presentation, discussion and conclusions. Minor editing is needed to improve English.
We hope the revised version of our manuscript now meets this reviewer’s expectations. We checked the language as thoroughly as possible and made several modifications to improve it as much as possible.